# Primary Care of the (Near) Future: Exploring the Contribution of Digitalization and Remote Care Technologies through a Case Study

**DOI:** 10.3390/healthcare11152147

**Published:** 2023-07-27

**Authors:** Federico Pennestrì, Giuseppe Banfi

**Affiliations:** 1IRCCS Istituto Ortopedico Galeazzi, Via Belgioioso 4, 20161 Milan, Italy; banfi.giuseppe@hsr.it; 2School of Medicine, Vita-Salute San Raffaele University, Via Olgettina 58, 20132 Milan, Italy

**Keywords:** algorithms, artificial intelligence, big data, informatics, NextGeneration EU, PNRR, primary care, telemedicine

## Abstract

The Italian Government planned to invest €15 billion of European funds on National Health Service digitalization and primary care enhancement. The critical burden brought by the pandemic upon hospital care mean these investments could no longer be delayed, considering the extraordinary backlogs of many treatments and the ordinary gaps of fragmented long-term care, in Italy and abroad. National guidelines have been published to standardize interventions across the Italian regions, and telemedicine is frequently mentioned as a key innovation to achieve both goals. The professional resources needed to run the facilities introduced in primary care are defined with great precision, but no details are given on how digitalization and remote care technologies must be implemented in this context. Building on this policy case, this paper focuses on what contribution digitalization and telemedicine can offer to specific primary care innovations, drawing from implemented technology-driven policies which may support the effective stratification, prevention and management of chronic patient needs, including anticipatory healthcare, population health management, adjusted clinical groups, chronic care management, quality and outcomes frameworks, patient-reported outcomes and patient-reported experience. All these policies can benefit significantly from digitalization and remote care technology, provided that some risks and limitations are considered by design.

## 1. Introduction

Next Generation EU is the European Commission’s €800 billion temporary recovery instrument to support the economic recovery from the coronavirus pandemic, and build a “greener, more digital and more resilient future” [1]. In April 2021, the Italian Government launched a National Recovery and Resilience Plan (in Italian: Piano Nazionale di Ripresa e Resilienza, PNRR) to invest about €190 billion of European funds in the six areas of digital innovation, ecological transition, mobility infrastructures, teaching and research, social cohesion and healthcare. Approximately €15 billion is dedicated to the latter (Mission 6), in particular to primary care enhancement (€7 billion) and the digitalization of the Italian National Health Service (NHS, approximately €8 billion). Telemedicine is frequently mentioned as a key innovation to achieve both goals [2].

The adoption of telemedicine, eHealth and information technology (IT) in support of primary care reform and chronic patient management has been recommended for a decade by the Ministry of Health [3,4] and by regional policy documents [5,6,7,8], although generally not described in detail, i.e., which technologies should be adopted to achieve which specific goals. Following the opportunity offered by NextGeneration EU and PNRR, the Ministry of Health published a Decree containing national guidelines on primary care modernization [9], binding on each Italian region and defining:the facilities that every region must adopt;the human resources needed to run each type of facility;the population that each facility is expected to cover;where telemedicine and digitalization should fit in.

However, (a) the stratification of patient needs on which primary care reform should be settled is based on algorithms and information systems described in previous policy documents that have not been implemented [10,11]; (b) no details are provided on how digitalization and telemedicine can support the “holistic evaluation of the health and social care needs of patients” [9] (p. 17) in terms of integrated care management.

The critical burden brought by the pandemic upon hospital care mean these investments can no longer be delayed [12], considering the extraordinary backlogs of many treatments and the ordinary gaps of fragmented long-term care, not only in Italy [13,14], but in all countries struggling to cover the needs of patients suffering from chronic morbidities [15,16,17,18,19,20,21,22] and social isolation [23]. For these patients, reducing distances is fundamental to maintain an acceptable quality of life, minimize life-threatening events and expensive preventable health complications, not to mention the direct burden on caregivers [24].

Building on this policy case, this communication paper focuses on what contribution telemedicine and remote care technology can offer to specific primary care innovations, synthesize and discuss them, both in terms of proactive governance of the healthcare demand (population health management) and integrated chronic patient management (planning, delivery, remuneration, evaluation).

The argument is structured by:Abstracting from national guidelines the key areas, functions and services in which digitalization (IT, integrated platforms, electronic clinical records, big data, artificial intelligence etc.) and remote care technologies (telemedicine, teleconsultation, televisit, telereferral, telemonitoring, telerehabilitation etc.) can make a contribution to primary care modernization;Describing some technology-driven policies which may support the effective stratification, prevention and management of chronic patient needs, integrating and/or filling the gaps of national guidelines, inspired by anticipatory healthcare, population health management, adjusted clinical groups, chronic care management, quality and outcomes frameworks, patient-reported outcome measurements (PROMs) and patient-reported experience measurements (PREMs);Pointing out some possible limitations, red flags and risks which are worth-considering by design.

We believe that this policy case can interest the multidisciplinary healthcare community (researchers, care professionals, policy makers, users) of all the systems called to handle the pressure of aging and chronic disease under limited financial resources, in order to share ideas, solutions, criticalities and ultimately help accelerate the reform of primary care.

## 2. Telemedicine and Primary Care Facilities

National guidelines have been published in Italy to standardize healthcare provision across public and private providers [25], hospital care [26] and ultimately primary care [9]. The latter are divided into 16 chapters among which No. 3 illustrates the stratification of patient needs on which the entire system should be based, No. 15 is entirely dedicated to telemedicine, and the eleven chapters in between introduce several interconnected primary and intermediate care facilities, all of which may benefit from telemedicine.

Telemedicine is defined as “A method of providing healthcare and long-term care remotely, enabled by information and communication technologies, and used by healthcare professionals to care for patients (professional—patient telemedicine), offer advice and support other professionals (professional—professional telemedicine)” [9] (p. 53). It consists of patient specialised televisit, teleassistance, telemonitoring, telerehabilitation, teleconsultation and telereporting, for the purpose of:Reducing physical distance between (a) healthcare professionals and patients, (b) healthcare professionals themselves;Connecting multiple healthcare professionals and providers, especially where integrated, multidisciplinary patient management is required (i.e., patients affected by chronic disease and/or social vulnerability, such as living alone, with limited autonomy and/or cognitive function);Performing early diagnoses and timely interventions in case of occasional acute events affecting chronic patients assisted at home;Providing efficient coverage and greater proximity to underserved areas, incrementing appropriate care and therapeutic compliance;Connecting primary and secondary care providers to promote de-hospitalization with safe patient discharge.

In this sense, telemedicine is considered a key driver to increase patient-oriented care and the responsive capacity of the NHS [9] (p. 54), reducing care gaps and inequalities caused by the social determinants of health (i.e., living place, distance from care facilities, family composition, income, health literacy, type of dwelling) which make certain population groups more or less consciously vulnerable (i.e., elderly chronic patients and/or patients with severe physical impairment and/or mental disability). That is why interconnected primary care facilities can benefit from telemedicine in several ways. In Table 1 these facilities are synthesized and divided per main function and services according to the guidelines provided by the Ministry, by the authors, in order to highlight how digital and remote care technologies can generate additional value in each area, according to the authors.

## 3. The Role of Technology in Population Health Management and Integrated Care Management

Not only can digital and remote care technologies fit in the intermediate and primary care facilities introduced by national guidelines. These technologies are likely to provide a substantial contribution to the proactive governance of healthcare demand and the integrated management of chronic patients.

### 3.1. Proactive Governance of Healthcare Demand

By proactive governance of healthcare demand, we refer to anticipatory approaches in which a specific patient population (i.e., national, regional, local) is stratified into homogeneous care clusters, associated by:Prevailing clinical needs (i.e., index disease and presence of comorbidities);Average care consumption (i.e., drugs, outpatient visits, hospital admissions and possibly social policies of healthcare significance, such as social prescribing, reablement, formal care giving or out-of-pocket payment exemptions) [27,28,29,30];Main area of intervention (i.e., hospital, primary, home or residential care facilities),

based on integrated administrative workflows provided by a common authority and shared by providers. The anticipatory approach lies in taking charge of patients before they develop or exacerbate certain disease conditions, or control them outside hospitals, both in terms of primary prevention (i.e., lifestyle education by GP advice), secondary prevention (i.e., screening programs and diagnoses) and tertiary prevention (minimizing acutization and complications through single-patient case management). 

The result of stratification is a set of adjusted clinical groups meant to advice the best approach to assist an individual patient or group of patients (e.g., case management, disease management, self-care education), the appropriate organization of care (e.g., setting and integration), targeted policy interventions (e.g., dehospitalization, public health campaigns, integration with social care) and the subsequent allocation of resources. While the concept of adjusted clinical groups dates back to the work of Barbara Starfield and colleagues at The Johns Hopkins University in the late 1970s [31], population health management was systematically developed more recently by Kaiser Permanente, a non-profit organization covering almost 13 million American patients, in a different way from traditional insurance methods in the United States [32] (Figure 1).

Thanks to the continuous development of IT, these models gain greater impact today. Adjusted clinical groups are defined by algorithms that work on a large number of codified data, drawing information from different providers, integrating such information into single aggregated outputs and making possible the holistic assessment before mentioned (e.g., single patient visual history of care, average healthcare consumption, costs and current estimated need). Details on these algorithms and their applications are described by technical guidelines in support of the policy documents released by the Italian regions that invested in this approach [5] (pp. 58–61), [7] (p. 31), [34] (pp. 22–23).

Integrated information systems constantly update patient stratification on a common database, building on electronic health records shared on a regional [35], NHS [36] or public insurance base [37]. Electronic records enable patients to (a) check the progress of their medical history on an integrated dashboard, finding medical reports, diagnostic exams, lab tests, forthcoming appointments and certifications (e.g., vaccinations, sports); (b) add information directly from home, thanks to wearable devices, web applications, patient-reported outcomes, including participation to remote decentralized clinical trials [38].

The region of Lombardy adopted some of these technologies in a radical primary care reform effort, from 2015 onwards, interrupted by the violent outbreak of the COVID pandemic and then superseded by the national guidelines discussed here. The approach, broadly comparable to that proposed by the regions of Emilia Romagna, Sardinia, Tuscany and Veneto [5], offers important advice to support and complete the implementation of national guidelines on primary care modernization.

The region first introduced an integrated patient DataWareHouse, in 2002 (Banca Dati Assistito, BDA), including the health-related information useful to track the patient history of care across the providers where he or she was assisted, monitoring process and disease outcomes [34] (pp. 18–21). After a dedicated algorithm was developed and two pilot experimentations were run [39], a population health stratification similar to Kaiser Permanente’s was adopted, in 2017, to define five levels of resource-intensive care need (Table 2), and drive the proactive recruitment of patients in dedicated care pathways of varying complexity.

Then, each patient belonging to Level 1–3 was assigned an index pathology (e.g., diabetes) divided by three degrees of estimated complexity, based on eventual comorbidities (e.g., diabetes and heart failure), average healthcare consumption per year (e.g., outpatient drugs and visits), hospitalization (based on the Diagnosis Related Groups system, DRG), social need indicators (e.g., income, family composition, instruction) and patient-share exemptions, namely exemptions from outpatient treatment payment associated with specific chronic diseases. These data were provided by the integrated patient database and aggregated by algorithms. 65 index pathologies were run first, while social vulnerability indicators were being developed. An example reporting five index conditions is represented in Table 3.

From the general patient stratification useful for epidemiological investigation and macro allocation of resources, each single patient affected by a chronic disease was proactively invited to contact the most appropriate care manager to meet their need, (a) evaluating the effective need in-person, (b) sharing a sustainable care plan (i.e., a plan that the patient is able and/or willing to comply with), (c) eventually taking charge of the patient.

### 3.2. Integrated Care Management

Each identified chronic patient was proactively contacted by the region with a paper letter and/or GP advice, and proposed a dedicated care manager able to provide (a) medical care for the index disease (i.e., with a general practitioner or specialist), (b) all the complementary treatments needed by the patient, either directly (in a single outpatient facility) or indirectly (by prescribing inpatient care or delegating services to connected third party suppliers). The idea was to evolve from a fragmented, disease-oriented, single “silos” approach, in which the patient is required to co-ordinate all the treatments, facilities and care professionals on their own (Figure 2), to an integrated, patient-oriented pathway digitally registered on a regional platform, clinically co-ordinated by a clinical manager, organizationally co-ordinated by a case manager and continuously updated with relevant information (e.g., test referrals, prescriptions, hospital clinical records, medical reports, exemptions) (Figure 3).

In this model, digitalization and remote care technology offer several opportunities (Table 4).

### 3.3. Compatibility and Scalability

Population health management is compatible with the stratification model proposed by the Ministry of Health [9] (p. 19),

(1)offering an already structured solution on which specific digital and remote care technologies can be implemented,(2)based on policies largely shared by regions already working on primary care modernization,(3)stressing the need to provide a unified information system on a national scale [9] (pp. 56,72),

to accelerate standardization and facilitate data exchange.

The sooner the data are unified and standardized, the greater scientific opportunities open up nationally and internationally, as big data science, artificial intelligence and machine learning will offer growing insights into epidemiology, policy, technology advancements and healthcare progress more in general (Figure 4), including outcome identification and systematic collection to express the best care value for patients (see next section). 

This approach is also consistent with the primary and intermediate care facilities introduced by the Ministry of Health, as care managers are easy to overlap with Case di Comunità (and previous Case della Salute, for example, in the Region of Tuscany) [8,43], clinical managers can work in Case di Comunità and case managers can work in Case di Comunità and Centrali Operative Territoriali.

However, the key step is to have clinical managers, care professionals with coordination skills and integrated primary care networks as close as possible to the home of chronic patients, and able to ensure continuous, financially sustainable, integrated patient-centered care regardless of the specific name and formal office they are assigned.

### 3.4. Further Advancement Opportunities

Once a solid infrastructure for patient stratification and integrated care management is settled, further opportunities are offered by digitalization and remote care technology to strengthen primary care.

A first one is agreeing additional remuneration for doctors who support prevention, patient empowerment, self-care and health education. IT allows the collection of primary care outcomes through digital platforms connected with supervising agencies and the funder, making systematic monitoring and rereward possible. An example is the Quality and Outcomes Framework introduced by the British NHS in 2004 [44]. According to the program, GP contract negotiations every year are based on a set of indicators which measures some relevant public health improvements to which they have contributed. Doctors who help patients comply with therapy or reduce harmful lifestyle habits receive additional remuneration. The indicators are reported on dashboards digitally connected with the NHS and generally cover:the management of spread chronic conditions such as asthma and diabetes;the management of major public health concerns such as smoking and obesity;the adherence to preventative screening programs or blood pressure checks [39].

The Framework increased the impact of prevention and reduced health inequalities among disadvantaged cohorts of the population [45,46], though NHS controls were needed to minimize GP gaming [47]. Pilot experimentations on breast cancer screening in the Italian region of Tuscany confirmed these benefits [48].

Beyond individual prescriptions and advice, clinical managers and GPs can also be incentivized to coordinate the entire care pathway by recomposing the various functions (professionals, facilities and treatments) in the patient’s interest. Several solutions can be implemented for this purpose, with more or less strong ties to the outcomes achieved.

Bundled payments, for instance, are (different versions of) pay-for-coordination solutions in which integrated healthcare pathways are remunerated when a clear set of pre-established outcomes is met [49,50]. The aims are (a) to clarify the outcomes that make a difference in the treatment of a certain patient or condition; (b) to design patient-oriented care pathways addressed to meet these specific outcomes; (c) to bound co-ordination on effective patient health improvement. Integrated IT and digital patient platforms are key to support this mechanism transparently and efficiently, as they are necessary to track each treatment provided, set and update goals and eventually validate remuneration (including eventual reasons to justify deviations). 

Personal health budgets are another pay-for-coordination solution launched by the British NHS in 2009, in which patients are given a money sum to pay for all treatments associated with a specific health condition based on an individual pathway, agreed by the NHS and provided by public or private-accredited care professionals and facilities [51]. The budget is credited (a) directly to the patient or tutor, (b) to NHS providers, (c) to a care manager able to provide all treatments included, comprehensive of eventual social prescriptions of healthcare significance. Funding is not tied to outcomes achievement although digital data collection is mandatory to check for the appropriate use of funds [52]. A similar solution was adopted by the region of Lombardy between 2012 and 2013 with chronic-related groups, a pilot experimentation with comprehensive care budgets to treat more than 60,000 patients affected by diabetes, heart failure, chronic obstructive pulmonary disease and hypertension. The patients who underwent the program reduced the frequency of emergency department visits and inpatient admissions in comparison to the control group (patients who were normally assisted). Later on, clinical managers and GPs were given additional remuneration based on the complexity (L1, L2, L3) of patients they had in charge (see Table 2 and Table 3) [14].

Setting clear outcomes by design is key to incrementing the value of integrated care pathways whether or not they are tied to provider remuneration [53]. Not only because clear goals are fundamental to address the coordination of care in a multiprofessional and multidisciplinary context; more in general, setting outcomes in advance makes it possible (a) to define the specific improvements expected, (b) to focus on what is most relevant and sustainable for a certain patient, enabling shared decision-making and facilitating compliance, (c) to reduce wasteful expenditure as a consequence; according to the words of the European Commission Expert Panel on effective ways of investing in health, to produce personal, technical and allocative value [54].

Patient-reported evaluations should be included for this purpose. The collection of PROMs in dedicated registries help grasp the patient perception on specific diseases, evaluating the benefits of different treatments, on different patients, and at different time stages [55]. Therefore, they support evidence-based clinical and allocative decisions. At the individual level, clinical managers can keep track of patients’ conditions in relation to certain treatments, therapy adjustments or pathway deviations, even more by using telemedicine devices to monitor patient parameters on a connected platform. On an aggregate level, the digital collection of aggregated data on electronic registries help physicians figure out what treatments work best on which type of patient, and show them the expected benefits, relapses and relative time frames when choosing the best plan based on shared, informed decisions [56]. Funders can also provide additional remuneration to providers who report better patient-reported outcomes for the same service [57]. Finally, the collection of PROMs is employed to estimate the quality-adjusted life years gained after chronic care treatments or elective surgery interventions, in order to capture the best value for different patients [24] and/or compare the performance of different providers at the international level: for instance, an Organization for Economic Cooperation and Development (OECD) evaluation on the benefits of hip and knee replacement demonstrated an average gain of 4.3 and 3.3 years in full health respectively including a specific Italian hospital (IRCCS Istituto Ortopedico Galeazzi) and seven world areas (Australia, England, Netherlands, Sweden, Switzerland, and the Canadian regions of Alberta and Manitoba) [58].

At the big data level, the digital collection of standard questionnaires can also discover unexpected correlations between patient characteristics (e.g., age, sex, comorbidities, ethnicity) and treatment outcomes (i.e., more or less beneficial, at what stage, to what extent), when supported by big data science and machine learning algorithms. Then, these correlations offer valuable research inputs to understand whether some treatments are more or less effective in certain patients, distinguishing clinical reasons (biomedical research) from other (policy and service management research). On a local level, for instance, outcome inequalities are often the consequence of implicit physical barriers, technology bugs, fragmented care pathways, difficult service access, poor therapy compliance, patients unable to understand advice, clinicians unwilling to explain treatments plainly or answer patient questions in a way they can understand [16,24]. Then, the collection of PREMs helps to investigate the environmental and relational factors which hamper or facilitate the achievement of best possible health outcomes, identifying room for improvement in terms of care organization and/or professional skills.

If patient questionnaires are collected and integrated on platforms, they support efficient clinical and policy assessments. In chronic care management, organizational and policy research can make a difference as much as clinical and biomedical research, or more: otherwise, the risk is to have individually excellent professionals and increasing health expenditure wasted by poor coordination and management [59]. IT, big data and machine learning innovation can offer important evidence and enormous room for improvement here.

## 4. Limitations

The adoption of digital and remote care technologies is not free from risks and limitations. Here is a list without the ambition of being exhaustive.

Creating a national database containing so much information on patients and providers is challenging. To be realistic, this step must be pursued gradually. Having uniform, ready-to-use regional databases should be preparatory. Care managers and/or primary care facilities should in turn have implemented compatible software themselves. Different countries and regions start at different levels right now, probably not only in Italy. Institutional co-ordination and strong leadership are fundamental to accelerate standardization.Cybersecurity and sensitive data protection are notoriously key issues in the contemporary technical, legal and ethical debate on technology regulation, with particular concern to artificial intelligence and machine-learning applications in healthcare [60,61,62]. Potential harm may follow from the use of data for purposes other than those for which they were collected and authorized (e.g., commercial use, privacy violation, or disease-related discrimination, including job loss). The same Hippocratic commitment to “first, not harm” can be jeopardized [63]. The more people have access to data, the more the risk of leaks, whether intentional or not. The consequences may become even more critical in market-driven healthcare systems, where chronic patients or patients at risk can be denied coverage. A central challenge here is to balance data safety and confidentiality on the one side and platform flexibility, (inter)accessibility and efficiency on the other.Liability for technology failure is another key issue. Machines will not replace doctors but likely support their ordinary diagnostic, prognostic and decision-making activity. Who is responsible for wrong decisions and prescriptions when they cause serious or irreversible harm to patients? Are patients able to discern the superficiality of a physician from the error of the machine he or she uses, eventually mandatory? Are they even interested in such a distinction, if they are considering suing? Intentions and responsibilities are often given moral relevance much more than facts themselves.Doctors may take wrong decisions from wrong inputs previously given by biased colleagues and researchers exactly such as reading Galen or contemporary scientific papers, as the history of medicine clearly demonstrates. It is one thing if this is an isolated contingency, but something else if relying on machines capable of reproducing errors at much greater levels, mostly unconsciously. Loop thinking and spurious correlations are two examples [60] (pp. 16–17), [64], as most contemporary machine-learning prediction models are based on correlation and not on causation [65]. Think of a (near) future in which medical students will entirely learn to rely on algorithms and machines, without critical attitude, fundamental logical skills and individual experience. Such a risk must be contained by firmly maintaining logic, ethics, medical history and philosophy of science in graduating curricula.Loop thinking can also hamper scientific progress, which is a first cultural reason why machine learning should be integrated with traditional learning techniques, at least for now. A second cultural reason which may slow down the spread and effectiveness of digitalization and telemedicine can be the digital divide, both from patients unfriendly with digital technology (e.g., elderly patients, patients with disabilities or cognitive dysfunction) and care professionals unwilling to change their habits. These barriers are likely to disappear within generations. A third cultural reason can be the poor propensity to share data and work in groups, as needed by integrated care pathways and possible gainsharing policies. In case of integrated care pathways, the radical spread of chronic diseases requires indispensable efforts to change the culture of fragmentation and task-oriented care. Undergraduate training should be evaluated for this purpose [66] (pp. 100–102). In case of gainsharing, outcome-driven remuneration should be considered with caution, not to add further pressure on care professions and make them unattractive to next generations.The digital divide can also affect the environment in which professionals work and patients are assisted. A reliable internet connection is needed to avoid ordinary bugs in the practice of telemedicine (e.g., freezing videocalls or missing disease-related patient parameters monitored real-time at home). At the facility level, the more we lean on technology, the more we have problems if the system breaks down. Therefore, not only should the physical environment be ready to provide safe and efficient use of technology. Alternative routes should be maintained at least for emergency situations, with users aware of the phone number they should call and someone actually available to answer their concerns on time.Pay-for-performance, value-based schemes and outcome-driven remuneration should not end up making some patients and diseases overlooked because of cherry-picking drifts, i.e., because they are too difficult to treat, because they are affected by concomitating irreversible conditions or because health status progression is hard to measure [67]. In value-based care, “value” does not mean profitable patients, but relevant improvement for real-world patients, be it functional recovery, regression slowdown or pain control (i.e., palliative care), at sustainable costs by the payor and the community (which are equivalent in an NHS). Providers, of course, need to generate value in turn, in terms of remunerated work, but strong regulation must be enforced in order not to sacrifice patient health to profit.A final important issue is finding the human resources willing to work in primary care facilities, as national guidelines are very specific about it. Hospitals can be professionally more attractive than primary care facilities, and care professional shortage, especially nursing, is an issue of global concern [68,69], considering the central role of nurses in primary care enhancement. However, this is not a reason to give up on strengthening primary care, a vital strategy for all the reasons mentioned before. This problem goes beyond the focus on technology, unless care professionals are replaced by robots.

## 5. Conclusions

Ten years of healthcare technology development can turn the Covid pandemic from a disaster into an opportunity, as technology had the chance to spread its ordinary use [70]. Now that the emergency phase of the pandemic has been declared closed, digitalization and remote care technology can be used to preserve the quality and sustainability of universal health coverage over time, especially in the sensitive area of chronic patients and long-term care, where hospital care is frequently not appropriate and is still unable to bear the pressure.

NHS maintenance and primary care modernization are required to grasp the NextGeneration EU opportunity on time, and technology offers significant potential at multiple levels. Red flags should be considered by design, in order to rely on solid infrastructures rather than running for cover after huge investments have been made. Some limitations are probably intrinsic to the degree of technology as it is today, and may improve with technical improvements. Other can disappear within a couple of generations as they are due to demographic trends, professional culture and training. Others bring ethical, legal and safety concerns which cross technology development more generally.

Let us assume that all of this works: primary care will by considerably strengthened and the population that was previously underserved will be efficiently taken care of, even before they come to the hospital with an acute complication or emergency. The final question, on which we provocatively want to finish, is whether full coverage will itself be sustainable in aging countries. This question is crucial for an equitable allocation of limited resources and requires careful reflection on the breadth of health coverage these countries will be able to afford and justify in the coming decades.

## Figures and Tables

**Figure 1 healthcare-11-02147-f001:**
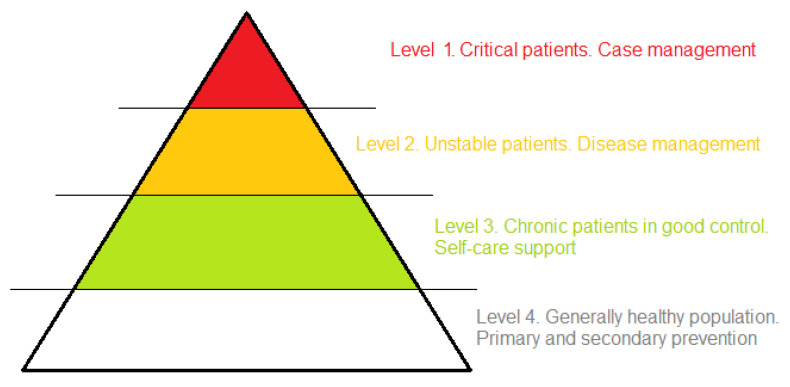
Population health management. Four-level model adopted by Kaiser Permanente, elaborated by the authors from [33] (p. 14).

**Figure 2 healthcare-11-02147-f002:**
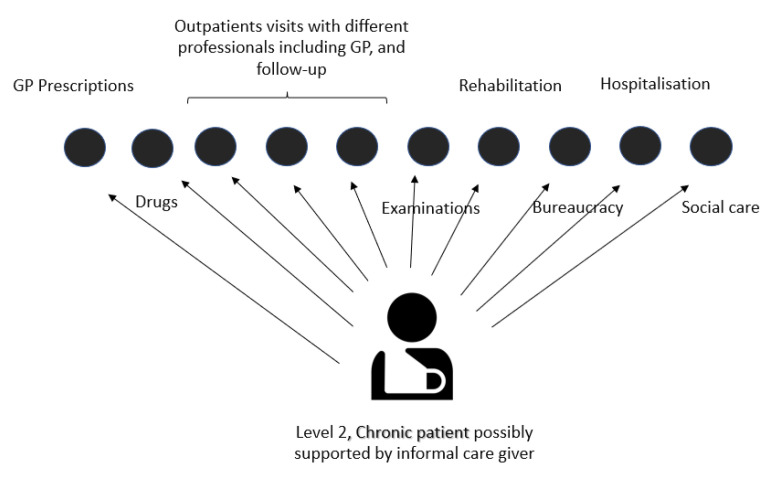
Fragmented care representation. Elaborated by the authors from [42].

**Figure 3 healthcare-11-02147-f003:**
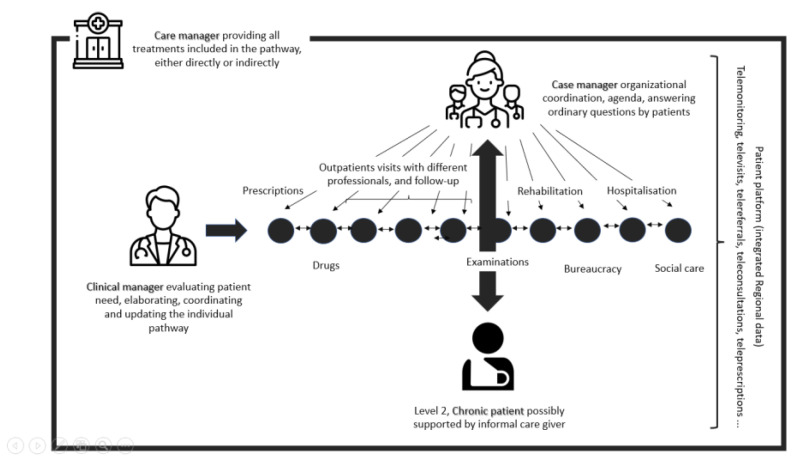
Digitally integrated patient management and remote care technology provision. Elaborated by the authors. See acknowledgements for graphic icons credit.

**Figure 4 healthcare-11-02147-f004:**
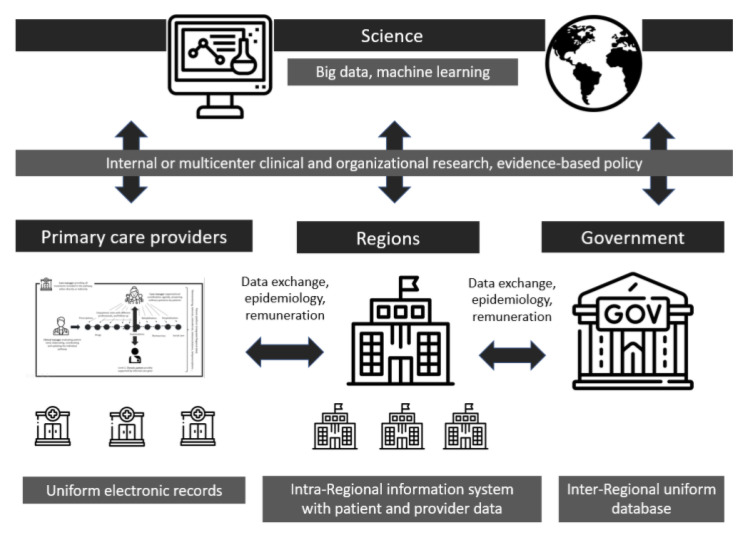
Model scalability. Elaborated by the authors. See acknowledgements for graphic icons credit.

**Table 1 healthcare-11-02147-t001:** Primary care facilities, main function and services which may benefit from digital and remote care technology.

Facility	Main Functions	Service	Technology
Casa di Comunità *. (Community Care Unit).	Outpatient clinic where General Practitioners (GPs) visit patients and perform basic diagnostics, multidimensional evaluation; connections with social care; maternity and childcare; ordinary specialistic visits are provided for high-prevalence chronic diseases; centralized administrative tasks (i.e., hospital reservations, drug and secondary care prescriptions, GP choice and revocation, medical certifications, payment exemptions are recognized).	Basic diagnostics.	Televisit, teleconsultation, teleremonitoring, telereferral related to ordinary chronic care conditions in patients at home. Teleprescription of ordinary drugs. Telereferral for driving licence renovation.
		Population screening.	IT can enable integrated platforms to inform patients on screening opportunities, where appropriate (i.e., age, previous relevant events). Integrated platforms can provide big data to enable population health management (see next paragraph). Artificial intelligence can work on big data to identify predictive factors, risks and unexpected disease determinants.
		Home care management.	Telemonitoring, telerehabilitation, informatics (see “Assistenza Domiciliare”).
		Central reservation system.	IT can enable integrated platforms to remind patients/care givers/case managers (see next paragraph) about upcoming appointments, either by phone, emails or message texts (i.e., outpatient visits, planned admissions). Integrated platforms can provide big data to enable population health management. Artificial intelligence can work on big data provided by integrated platforms to identify correlations between medical procedures and complications or side-effects (i.e., surgical infections, incompatible drugs, medium- and long-term side-effects).
Unità di Continuità Assistenziale. (Care Continuity Unit).	Mobile unit (i.e., doctor and nurse) ensuring the continuity of care in case of difficult contingencies (i.e., infectious outbreaks or logistical complications). Physically based in a Casa di Comunità.	Remote consultation with GP or specialist doctor.	Televisit, teleconsultation, telereporting disease parameters and patient outcomes.
		Telemonitoring in case of individual infection or local outbreak.	Televisit, teleconsultation, telereporting, telemonitoring.
		Home vaccinations.	IT can track patient compliance in support of collective immunization strategies, and trigger one-off information policies in case of non-compliance. IT can enable integrated platforms to support population health management (i.e., vaccine uptake). Integrated platforms can provide big data to identify correlations between patient characteristics (i.e., sex and age), comorbidities, vaccine effectiveness, safety or side-effects, even supported by artificial intelligence.
Centrale Operativa Territoriale. (Local operation center unit).	Nurse-led operating center ensuring links among primary care, hospital care, emergency care, connected to each individual providers’ software.	Patient tracking between transitions.	IT can enable integrated platforms to notify whether the patient have successfully been admitted/discharged/moved to the appropriate care setting, even more when non-autonomous (i.e., cognitive dysfunction or social isolation), to activate eventual arrangements on time (i.e., care giver, home care, wearable devices).
		Logistical and informative support.	IT can record effective or ineffective transitions and provide evidence for policy arrangements.
		Clinical data collection including home care.	See “Assistenza domiciliare”.
Assistenza domiciliare. (Formal home care).	Home-delivered care for non-autonomous patients waiting to stabilize clinical status, limit functional decline, improve quality of life, based on different levels of complexity evaluated by the GP and managed in a Casa di Comunità.	Activity records	Telemonitoring and telerehabilitation can help patients receive advice and stay safer at home. IT, big data, artificial intelligence can provide integrated platforms with health information collected at home, enhancing health policy, population health management and reducing the frequency of underreported health-related events such as side-effects or complications. Digital applications and wearable devices can track the health-related activity of patients at home, providing evidence-based information to evaluate compliance, performance variations and eventually alerting on time whether no physical activity or vital functions are registered. Similar devices can support decentralized clinical trials to increase trial retention while reducing travel to facility and eventually costs.
Ospedale di Comunità. (Community hospital).	Intermediate inpatient care for patients temporary unable to stay home safely but still do not need resource-intensive hospitalization, or patients discharged from hospital in need of clinical stabilization. Closer to home than high-volume central hospitals. Maximum 20 beds and 30 days of stay.	Safe discharge	IT can enable integrated platforms support patient admission from home to intermediate hospital, from central hospital to intermediate hospital, and tracking. Integrated platforms can provide big data to identify correlations between disease and length of stay, acute care hospital treatment and post-acute hospital length of stay, intermediate care admissions/length of stay and reduced acute hospitalizations, complications and emergency care visits, proximity to admitted patients’ home and relative effectiveness in reducing acutization. Artificial intelligence can find unexpected correlations between any of these factors (i.e., stability of internet connection and compliance to therapy, effectiveness and safety).
		Patient education (digital technology and self-care)	Digital education under a protected environment, in order to prepare patients to the eventual use of telemedicine and reduce the risks of the digital divide. Ensuring that the patient have stable internet connection at home should be considered.

* These facilities are divided in Hub and Spoke Units, based on the amount of population they are expected to cover. Some requirements compulsory for Hub Units are only recommended for Spoke Units. Distinctions are not reported here for simplification. Each facility is named in Italian according to the original version contained in the decree. The English translation is made by the authors to give the reader better understanding.

**Table 2 healthcare-11-02147-t002:** 5-Levels model, integrated information systems, patient-database driven population health management adopted by the region of Lombardy, elaborated by the authors from [40] (p. 12).

Need	Appropriate Setting(s) and Service Design	Number and Percentage of Patients over the Population of Lombardy *
Level 1. Complex, chronic patients at high risk of acutization, generally affected by comorbidities, functional limitations and social vulnerability.	Intensive integration between hospital and home care.	Approx. 150,000 (1.5%).
Level 2. Chronic patients characterized by comorbidities, predominantly outpatient clinical need and low-to-moderate social vulnerability.	Outpatient care, self-care education, tertiary prevention, intermediate or primary care facilities (i.e., aggregated GPs).	Approx. 1,300,000 (13%).
Level 3. Patients affected by a single and/or early-stage chronic disease, with low-to-moderate outpatient needs and no critical social vulnerability.	Outpatient care, self-care education, single GP and primary care facility.	Approx. 1,900,000 (19%).
Level 4. Non-chronic patients characterized by occasional healthcare consumption.	Not of this policy concern.	Approx. 3,000,000 (30%).
Level 5. Patients with negligible healthcare consumption.		Approx. 3,500,000 (35%).

* The population of Lombardy makes up 1/6 of the entire Italian population, namely 10,027,602 inhabitants at 31 December 2019 [41]. Today the population of Lombardy has dropped below 10,000,000 inhabitants after two years of the COVID pandemic, which affected old and fragile individuals most [12].

**Table 3 healthcare-11-02147-t003:** Algorithm-driven adjusted clinical groups, adopted by the region of Lombardy, elaborated by the authors from [40] (pp. 25–26).

Chronic Index Disease	Outpatient Level 3	Outpatient Level 2	Outpatient Level 1	Outpatient Drugs Level 3	Outpatient Drugs Level 2	Outpatient Drugs Level 1	Hospitalisation (Single DRG)
Integrated provider and patient database							
Diabetes, type 1	€ 317	€ 419	€ 787	€ 883	€ 1001	€ 1565	€ 3000
Dementia	€ 302	€ 362	€ 450	€ 533	€ 718	€ 972	€ 7598
Multiple sclerosis	€ 1220	€ 1131	€ 1089	€ 96	€ 458	€ 1056	€ 13,689
Heart failure	€ 450	€ 587	€ 740	€ 593	€ 992	€ 1420	€ 9117
Active neoplasia	€ 1945	€ 1646	€ 1704	€ 375	€ 733	€ 1293	€ 9747

**Table 4 healthcare-11-02147-t004:** Digitalization and remote care technology contribution to integrated chronic care management.

	Function	Facility or Professional	Technology
**Care manager**	Provision of all the treatments needed by the patient, either directly (single facility) or indirectly (delegated proximal service and hospitalization, when/where appropriate)	Single primary care facility, or network of primary care facilities, either public or private accredited, connected with hospital care (i.e., Azienda Socio-Sanitaria Territoriale, ASST in case of public care manager).	Integrated provider and patient platform software: Input: access to health-related patient data available to the regional information system (national in perspective). Output: recording the treatments provided on the regional information system (national in perspective).
**Clinical manager**	Patient assessment. Individual pathway design shared with patient and/or care-giver. Monitoring and updating. Prescribing treatments. Clinical co-ordination.	Medical doctor working in outpatient setting. GP in case of Level 3 chronic patient. GP in possible cooperation with a specialist of the index disease in Level 2 chronic patient. Specialist of the index disease in Level 3 chronic patient.	Televisits, telereferrals, telemonitoring, teleprescriptions in case of patients with limited mobility (either temporary, either permanent) or chronic ordinary needs (i.e., renewing prescriptions, monitoring parameters). Teleconsultations with colleagues (i.e., GP and specialist(s)).
**Case manager**	Monitoring chronic patients’ ordinary parameters and providing support in case of need. Consultation with clinical manager when appropriate. Reminding patients about follow-up visits, diagnostic exams, therapy updates, certifications. Connecting providers (i.e., externalized or hospital) to ensure continuity of care and/or safe transitions.	Advanced nurse with co-ordination skills, working in outpatient setting.	Wearable and/or environmental devices to enable patient monitoring. Televisits with patients and teleconsultations with clinical manager. Where possible, automatic renovation of ordinary care prescriptions.

## Data Availability

No new data were created or analyzed in this study. Data sharing is not applicable to this article.

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
