# Peer review of "Primary Care of the (Near) Future: Exploring the Contribution of Digitalization and Remote Care Technologies through a Case Study"

_healthcare, 2023, doi:10.3390/healthcare11152147_

Round 1

Reviewer 1 Report

This paper is interesting and well-written. It can be published as it is.

I only have some minor comments.

1. NHS is used to describe the national system in Italy which is highly associated with the English NHS. Could this be specified by using "the Italian NHS" as you do on page 9, line 319 (the English NSH)?

2. It was a bit unclear if the different facilities and functions described in Table 1 were based on the proposed guidelines, if these were in the making or if these were established.

3. Table 1: What does Telereporting mean, is it patient reporting outcome measures transmitted electronically?

4. Is renewing prescriptions better than renovation prescriptions?

Author Response

Dear reviewer,

thank you very much for the useful comments.

We specified "Italian NHS" the first time it is used for the rest of the paper (point 1). We better specified what you suggest at points 2 and 3. We corrected according to your suggestion at point 4.

Our best regards, the authors.

Reviewer 2 Report

The authors analyse how digitalisation and telemedicine can contribute to specific primary care innovations. They focus on the added value of implemented technology-driven policies in supporting the effective stratification, prevention and management of chronic patient needs.

Anticipatory healthcare, population health management, clinical groups, chronic care management, quality and outcomes frameworks, patient-reported outcomes and patient-reported experience are addressed throughout the paper.

The paper is very well structured and the quality of the overall presentation is high. The reported tables provide the reader with a very useful overview of the healthcare facilities and professional managers highlighting their main functions and mapping them into the appropriate technology that may support the related service. The paper might be a good starting point for the e-health technological system designers and developers.

The facility column of table 1 should be translated in English.

I recommend this paper for publication.

Author Response

Dear reviewer,

thank you very much for the positive comments.

We translated Table 1, facility column in English according to your request.

Our best regards, the authors.

Reviewer 3 Report

Although the benefits of telemedicine are well-known, the application of this technology is limited. Most national health IT directives do not provide details on how digitalisation and remote care technologies should be implemented. This paper presents an analysis of contributions (and issues) that telemedicine and remote care technology may offer to primary care.

A minor limitation of this paper is its focus on the Italian situation. Issues and opportunities are shared among, mainly, western countries; but there could be relevant differences among them that could be pointed out. References to cases in other countries would improve the paper.

Minor comments:

1. Table 1 is too large occupying several pages. Consider to synthetize the content or explain it in detail in text. Abstracting from the Italian case to a more general one would be valuable.

2. Figure 1 top: level 3 > level 1

Author Response

Dear reviewer,

thank you very much for the useful comments.

Lines 51-59, references 15-21 (now 15-22) and reference 22 (now 23) contextualize the need of renovating primary care and reducing fragmentation by the use of technology on a global level, already including a Chinese reference to which we added on from South America. Moreover, previous reference 22 (now 23) is a systematic review with meta analysis including studies from all over the world, including 7 from Asia, 7 from Australia and 3 from other "multiple regions" (see Table 2, p. 231 of the paper).

We agree that Table 1 is large but we also believe its value comes from representing graphically a schematic overview of primary care facilities, services and possible technological innovation, helping the reader focus on each specific point. Abstracting from the Italian case to a more general one would not be possible here because the starting focus on which the entire paper builds is this specific policy case (to which, afterwards, broader international context is integrated).

Finally and thanks to your recommendation, we corrected "Level 1" in Figure 1.

Our best regards, the authors.

Reviewer 4 Report

The Italian Government has committed €15 billion of European funds to digitally transform the National Health Service and enhance primary care. This decision stems from the urgent need to address the overwhelming strain on hospitals caused by the pandemic, coupled with the existing deficiencies in fragmented long-term care both within Italy and abroad. To ensure consistent approaches, national guidelines have been established, with telemedicine emerging as a vital innovation to achieve the desired objectives. Although the professional resources required for primary care facilities have been clearly defined, there is a lack of specific guidance on the implementation of digitalization and remote care technologies within this context. Drawing on this policy case, this paper aims to explore the potential contributions of digitalization and telemedicine to specific primary care innovations. It highlights the importance of implemented technology-driven policies that can effectively address the stratification, prevention, and management of chronic patient needs, including anticipatory healthcare, population health management, adjusted clinical groups, chronic care management, quality and outcomes frameworks, patient-reported outcomes, and patient-reported experience. Successful implementation of these policies hinges on careful consideration of associated risks and limitations.

Comment:

The article under review presents a comprehensive analysis of various dimensions related to healthcare, but falls short in providing depth and exploration of key aspects. While social determinants of health are acknowledged as a crucial factor for population clustering, their impact is not thoroughly examined. Additionally, the development and usability of electronic health records in identifying special needs and improving health outcomes are briefly mentioned, without delving into specific methodologies or examples from existing literature. Furthermore, the utilization of big data and algorithms in analyzing special data lacks detailed exploration, such as specific types of algorithms employed and their applications. Overall, the article would benefit from further expansion and inclusion of in-depth analysis in these areas to provide a more comprehensive and impactful examination of the subject matter.

Author Response

Dear reviewer,

thank you very much for the useful comments.

We wrote ll. 120-124 to mention specific social determinants which impact on health outcomes and inequalities caused by care fragmentation, recalling and strengthening the previous bullets on the potential benefits of telemedicine in primary care (ll. 107-120).

We added three references published in the international literature about the use and potential of electronic health records in patient stratification and health outcome improvement (one from a regional level, one from a NHS, one from a social insurance level). We could not add more references as we have already reached a great number and we need to add other references to meet also some recommendations by the other reviewers. In lines 2017-224 we mentioned the specific electronic health record system used by the Region of Lombardy to strengthen its use to identify needs and improve outcomes (both health improvements and process outcomes).

We deeply searched the documents cited in the paper to specify which algorithms are applied on big databases to analyze specific data on patient characteristics, health conditions, average consumption and costs. The reader can now explore this information in detail without reporting it systematically within the main text.

We hope our integrations meet your comments.

Our best regards, the authors.

Round 2

Reviewer 4 Report

NA